# FLARE REMOVAL WITH VISUAL PROMPT

## ABSTRACT

Flare removal methods remove the streak, shimmer, and reflective flare in flare-corrupted images while preserving the light source. Recent deep learning methods focus on flare extraction and achieve promising results. They accomplish the task by either viewing the flare equals to the residual information between the flare-corrupted image and the flare-free image and generating the flare-free image through subtracting the extracted flare image or generating the flare-free image and the flare image simultaneously. However, due to the gap between the flare image and the residual information and handling flare extraction and clear image generation process simultaneously will give the network too much pressure and cannot fully utilize the extracted flare, these methods tend to generate images with severe artifacts. To alleviate such a phenomenon, we propose a model-agnostic pipeline named Prompt Inpainting Pipeline (PIP). Specifically, instead of viewing the gap between the flare-free and flare corrupted image as the flare or generating the flare-free image and flare image simultaneously, our prompt inpainting pipeline provides a novel perspective. We borrow the idea from inpainting methods and remove the flare by masking the polluted area and rewriting image details within. Unlike inpainting methods, we first extract multi-scale features of flare-corrupted images as a visual prompt and rewrite missing textures with the visual prompt since we find out that directly writing the missing details based on the remaining area hardly generates promising image details with sufficient semantic and high-frequency information. To verify the function of our pipeline, we conduct comprehensive experiments and demonstrate its superiority.

## 1 INTRODUCTION

Flare removal methods aim to remove the unwanted lens flare in input images while preserving the light source. In theory, an ideal camera should be able to converge all rays from a single point source to a single focal point. However, in reality, due to the sophisticated imaging system and dirt on the lens, lenses reflect and scatter light along unintended paths, leading to the appearance of flare that produces brightness in radial areas of images. The flare severely degrades the qualities of images and hinders downstream visual tasks (*e.g.*, semantic segmentation, depth estimation). The industry commonly adopts Anti-Reflection (AR) coatings to suppress the reflective flare phenomenon, which is costly and only optimized for specific wavelengths and angles of light and hardly handles the scattering flare that frequently happens on smartphone cameras. Therefore, how to remove flare in images has attracted plenty of attention from industry and academiaAsha et al. (2019); Vitoria & Ballester (2019); Feng et al. (2023); Sun et al. (2020); Wu et al. (2021); Dai et al. (2022).

Researchers have made great efforts in this field. By viewing the residual information between the flare-corrupted image and the flare-free image equals to the flare image, Wu et al. Wu et al. (2021), Dai et al. Dai et al. (2022), and Dai et al. Dai et al. (2023b) employ a pipeline which adopt existing image restoration methods Zamir et al. (2021); Chen et al. (2021); Wang et al. (2022); Zamir et al. (2022) to extract the flare image and generate flare-free image by subtracting the estimated flare image with flare-corrupted image. However, there exists an obvious gap between the residual image and the flare image, and artifacts often happen. Moreover, Dai et al. Dai et al. (2023a) propose another pipeline named Flare7k++ and generate the flare image and flare-free image simultaneously. Such a method avoids the loss caused by the gap between the residual information and the flare image, whereas gives the network too much pressure and cannot fully utilize the extracted flare which leads to annoying artifacts on generated images. Thus, how to efficiently remove flare from images without introducing artifacts becomes challenging.

In this paper, we propose a model-agnostic pipeline named **P**rompt **I**npainting **P**ipeline (PIP) which can be equipped with any image restoration methods. To eliminate generated artifacts, especially when handling images with strong streaks, we provide a novel perspective and separate the flare removal process into a coarse flare removal stage and an image refinement stage. The coarse flare removal stage adopts a U-shape network to coarsely generate the flare-removed image and the flare. As for the image refinement stage, we borrow the idea of inpainting methods, which mask the area polluted by flare and rewrite the missing details within. Nevertheless, instead of compelling the network to infer the missing pixels based on the remaining area, we propose prompt calibration block which adopts features extracted during the coarse flare removal stage as a visual prompt to guide the rewriting process and generate promising flare-free images. In this way, our PIP pipeline manages to erase the flare and decrease artifacts.

We conduct comprehensive experiments on the PIP pipeline and experimental results on real-world benchmarks prove the effectiveness of our method. Our contributions can be summarized as follows:

- We propose a model-agnostic pipeline named PIP which suppresses artifacts by rewriting details corrupted by flare.
- The proposed prompt calibration block adopts features extracted during the coarse flare removal stage as a visual prompt to guide the rewriting process.
- Extensive experiments demonstrate the superiority of our method.

## 2 RELATED WORK

Flare is an optical phenomenon, in which light is scattered and/or reflected in a sophisticated optical system. The flare may seriously degrade the image's quality and hinder downstream visual tasks. Thus, numerous efforts have been made to mitigate this issue.

### 2.1 HARDWARE SOLUTIONS

To capture high-quality images, modern digital optical systems tend to employ multiple glass elements. The increase of the glass elements raises the probability that light reflects from its surface and generates flare. To eliminate the reflective flare, most hardware solutions focus on improving the optical system (*e.g.*, sophisticated lens barrel design, lens hoods, reflective coating). Applying AR coating Nussberger et al. (2016); Raskar et al. (2008) to lens elements reduces internal reflection by destroying interference, which is widely adopted in practice. However, AR coatings are costly to add to all optical surfaces and the thickness of the coating is designed for specific wavelength and angle of incidence. Moreover, scattering flare always occurs when dust and fingerprints appear in front of the lens. Therefore, these efforts hardly eliminate the flare generated in the pre-process and cannot handle images with flare during the post-process.

### 2.2 SOFTWARE SOLUTIONS

Traditional flare removal methods Faulkner et al. (1989); Reinhard et al. (2010) have achieved good performance based on statistics prior. Since the development of deep learning, numerous methods have been proposed.

#### 2.2.1 DATASETS

Aiming for a low-level task, the performance of flare removal methods depends on the qualities of the datasets. A dataset that contains a large amount of data pairs can greatly improve the performance of neural networks when handling real-world scenarios.

To this end, Wu et al. Wu et al. (2021) propose a semi-synthetic dataset that contains 2001 captured flare images and 3000 simulated flare images. However, they focus on daylight flare and the proposed dataset tends to be less generalization to real-world flare which can be captured by diverse lenses and light sources, especially at nighttime. Therefore, Dai et al. Dai et al. (2022) propose a nighttime flare removal dataset Flare7K, which contains 5000 scattering and 2000 reflective flare images. Furthermore, Dai et al. [Dai et al. (2023a)] propose Flare7K++, which is an extended version of Flare7K. On the basis of Flare7K, Flare7K++ gives each image an additional light source

annotation and proposes a new real-captured subset Flare-R which contains 962 flare images. These datasets exhibit high sensitivity to scattering flares while giving insufficient attention to reflective flares. Hence, Dai et al. Dai et al. (2023b) propose the first reflective flare removal dataset named BracketFlare dataset based on the prior that the reflective flare and light source are always symmetrical around the lens's optical center. They employ continuous bracketing to capture the reflective flare pattern in unexposed images and aggregate with exposed images to synthesize paired data. They conduct experiments and prove that neural networks trained on this dataset gain the capability of removing the ghosting effect in images.

### 2.2.2 NETWORK STRUCTURE

As capturing large amounts of data pairs for flare removal is challenging and tedious, earlier deep learning methods He et al. He et al. (2010) tend to utilize unsupervised methods. Qiao et al. Qiao et al. (2021) propose a generative adversarial network-based learning framework to learn from unpaired data. They adopt the idea of cyclegan Zhu et al. (2017) and separately detect the light source region and the flare region. The output is generated by blending the flare-removed image and the detected light source mask. Their method achieves promising results when handling tiny light sources and flares, whereas fail on images with strong light sources and large flares.

As multiple synthetic and real flare removal datasets have been proposed, Wu et al. Wu et al. (2021) and Dai et al. Dai et al. (2022) adopt many end-to-end image restoration methods Ronneberger et al. (2015); Chen et al. (2021); Zamir et al. (2022); Wang et al. (2022) to extract the flare image from the input flare-corrupted image. They surpass the unsupervised methods, whereas still generate artifacts during the flare removal process as they view the gap between the flare-corrupted image and the flare-free image equal to the flare image. Meanwhile, Dai et al. Dai et al. (2023a) propose a different pipeline named flare7k++, which separately extracts the flare and restores the image in a simultaneous manner and adopts image restoration networks to do both jobs instead of simply generating the flare-free image. However, such a method compels the network to analyze the flare pattern and generate flare-free images simultaneously, which some artifacts still happen, especially when a strong streak appears. To this end, we propose a model-agnostic PIP pipeline to reduce the flare and generate images with fewer artifacts. By borrowing the idea of inpainting methods and separating the flare removal process into a coarse flare removal stage and an image refinement stage, our pipeline surpasses state-of-the-art methods.

## 3 METHODOLOGY

To better illustrate our PIP pipeline and distinguish it from the previous pipeline, we introduce the Flare7k++ pipeline first. Sequentially, we illustrate the details of our PIP pipeline and demonstrate its superiority.

### 3.1 FLARE7K++ PIPELINE

As shown in Figure 1 (a), the Flare7k++ pipeline adopts a network with a U-net backbone to predict the flare-free image and the flare image which excludes the light source information. In this way, the network can preserve the light source image and better locate the flare image by individually estimating the flare image. However, due to the pixel limit of digital systems, some information is blocked by overexposed streaks. Such degradation is hard to compensate for as the blocked area contains little useful information and the Flare7k++ pipeline requires the network to extract the flare and generate flare-free images simultaneously, thereby giving too much burden for the neural network and leading to severe artifacts. To this end, we propose our two-stage pipeline named PIP to reduce the artifacts.

### 3.2 PROMPT INPAINTING PIPELINE (OURS)

Our PIP pipeline provides a novel perspective for the flare removal task. Existing flare removal pipelines accomplish this task under the idea of image-to-image translation. However, such a design may be suboptimal for the flare removal task as the streak and shimmer in the flare-corrupted image occasionally appear to be strong and fully block the content embedded and these methods may

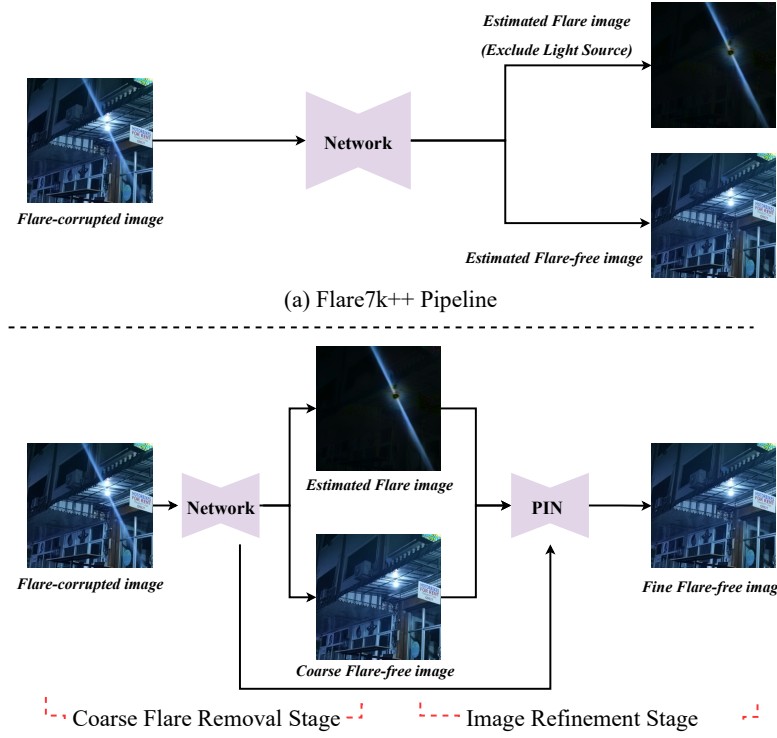

(a) Flare7k++ Pipeline

(b) Prompt Inpainting Pipeline (Ours)

Figure 1: Comparison between Flare7k++ pipeline Dai et al. (2023a) and our PIP pipeline.

generate severe artifacts when handling such flare. Therefore, we propose the PIP pipeline, which accomplishes this mission by rewriting the image details occupied by the flare.

Figure 2 shows the details of the PIP pipeline. As a coarse-to-fine two-stage pipeline, the PIP pipeline consists of a coarse flare removal stage and an image refinement stage. Concretely, our PIP pipeline is a model-agnostic pipeline. During the coarse flare removal stage, any U-net flare removal method can be employed as multi-scale features are useful in such image translation tasks, and most methods in this task adopt one U-net structure as the backbone to extract multi-scale features. As for the image refinement stage, we propose the Prompt Inpainting Network (PIN) and employ a U-net backbone under the guidance of the multi-scale features extracted during the first stage.

### 3.2.1 COARSE FLARE REMOVAL STAGE

In this stage, we coarsely remove the flare from the flare-corrupted image by estimating the coarse flare-corrupted image and the entire flare image. The loss function is formulated as:

$$\mathscr{L} = \begin{cases} \dfrac{1}{N} \sum_{i=1}^{N} |I_{FF}^{i,Coarse} - I_{gt}^{i}| \\ \\ \alpha \dfrac{1}{N} \sum_{i=1}^{N} |F_{DF}^{i} - F_{gt}^{i}| \end{cases} \tag{1}$$

where $I_{FF}^{i,Coarse}$, $F_{DF}^{i}$, and $F_{gt}^{i}$ represent the $i^{th}$ pixel in the output coarse flare-free image, the output flare image, and the ground truth image.

### 3.2.2 IMAGE REFINEMENT STAGE

After obtaining the flare image which is added on top of the flare-free image and the coarse flare-free image, we introduce the image refinement stage to further remove the flare and artifacts from the coarse flare-free image. We do not adopt an image-to-image translation network as we argue that such a design hardly handles severe degradation. In the image restoration task, simply adopting such

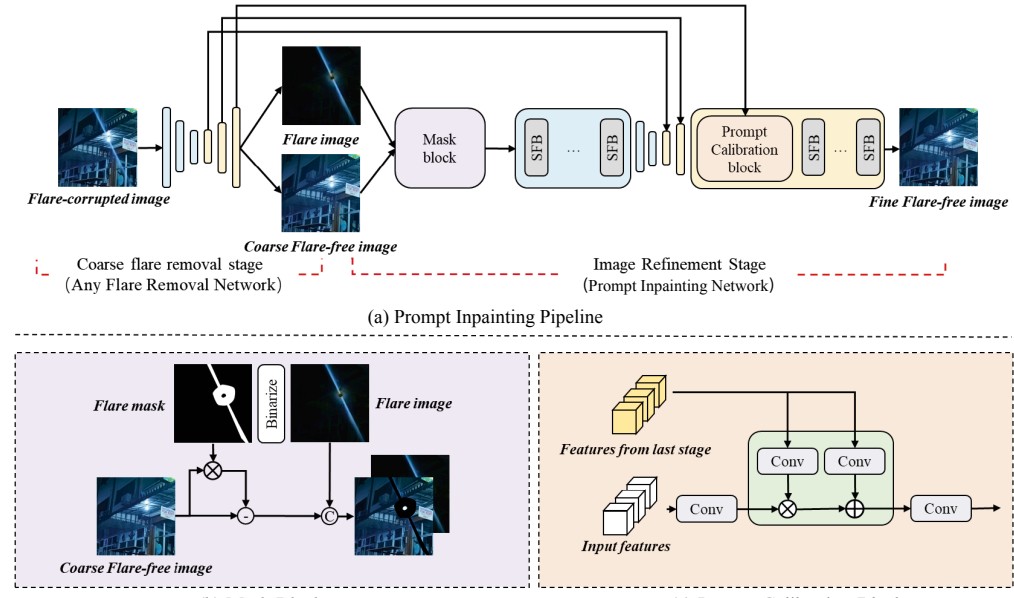

Figure 2: The overview of our PIP pipeline. Our PIP pipeline consists of a coarse flare removal stage and an image refinement stage. The coarse flare removal coarsely estimates the flare and removes the flare from the flare-corrupted image. The image refinement stage further removes the artifacts introduced during the coarse flare removal stage. We rewrite the missing details polluted by the flare image and employ a U-net structure as our backbone to eliminate the flare under the guidance of the multi-scale feature extracted from the last stage.

a network on image restoration suffers from unpromising results when handling severe degradation (*e.g.*, complicated motion blur, high-level noise). To this end, we accomplish this task from a novel view by extracting the semantic information from the unpolluted area, rewriting the details in the polluted area based on the extracted semantic information, and using multi-scale features extracted from the last stage as the visual prompt. Specifically, we propose the PIN network in this stage which adopts the mask block to generate the input and a U-net neural network to rewrite the missing details.

**The mask block**: The procedure of mask block is depicted in Figure 2 (b). Given the estimated flare image, we first binarize it to obtain the mask with a predefined threshold and multiply it with the coarse flare-free image $I_{FF}^{Coarse}$ to obtain the masked image $I_{FF}^{mask}$. We select the coarse-generated flare-free images instead of the flare-corrupted image as the estimated mask omits the flare with slight luminance and using the flare-corrupted image may leave these flare on the final output. Secondly, we subtract $I_{FF}^{mask}$ with $I_{FF}^{Coarse}$. Finally, we generate the input $I_{FF}^{Input}$ by concatenating the image obtained in the last phase with the estimated flare image $F_{DF}$. The process is formulated as:

$$Mask = \text{Binarize}(Mask > threshold), \qquad I_{FF}^{mask} = I_{FF}^{Coarse} \times Mask,$$
$$I_{FF}^{mask} = I_{FF}^{Coarse} - I_{FF}^{mask}, \qquad I_{FF}^{Input} = \text{Concat}(I_{FF}^{mask}, F_{DF}) \tag{2}$$

**U-shape Structure**: Given $I_{FF}^{Input}$ and features obtained from the last stage $X_{FF}^{coarse}$, we adopt a U-net structure for the inpainting task. Firstly, the encoder in the backbone employs 4 SFB blocks Zhang et al. (2023) at each level and extracts the multi-scale semantic feature of $I_{FF}^{Input}$. Thereafter, the decoder rewrites the missing details by using $X_{FF}^{coarse}$ as the visual prompt. Concretely, we adopt one prompt calibration block and 4 SFB blocks at each level in the decoder and the prompt calibration block is sequentially employed to utilize $X_{FF}^{coarse}$.

**The prompt calibration block**: The prompt calibration block refines the image features by employing the $X_{FF}^{coarse}$ as the visual prompt and the structure is depicted in Figure 2 (c). The formulation of the prompt calibration block is presented as follows:

$$X^i = \text{Conv}(X_{FF}^{i,Coarse}) \times \text{Conv}(X^i) + \text{Conv}(X_{FF}^{i,Coarse}) \tag{3}$$

where $i$ represents the i$^{th}$ level in decoder.

The loss function of this stage is the L1 distance and perceptual loss between the output and the ground truth flare-free image. The formulation is shown as follows:

$$\mathscr{L} = \begin{cases} \dfrac{1}{N} \sum_{i=1}^{N} |I_{FF}^{i,fine} - I_{gt}^{i}| \\ \mathscr{L}_{per}(I_{FF}^{i,fine}, I_{gt}^{i}) \end{cases} \tag{4}$$

where $I_{FF}^{i,fine}$ represents the output in refinement stage and $\mathscr{L}_{per}$ means the perceptual loss.

## 4 EXPERIMENTS

In this section, we conduct ablation experiments on modules proposed within the PIP pipeline on real-world data. Moreover, we conduct comparison experiments on the Flare7K++ real dataset Dai et al. (2023a) and BracetFlare dataset Dai et al. (2023b).

### 4.1 DATASETS

For the training set, we adopt Flare7K++ Dai et al. (2023a) and BracketFlare Dai et al. (2023b) datasets. Notably, Flare7K and Flare7K++ provide a quantity of flare images, while giving no flare-free images in the training set. We follow the experimental setting in Flare7K which adopts the Flickr24k dataset Zhang et al. (2018) as flare-free images. To form the data pair for supervised training, we add compound flare images on top of flare-free images to generate nighttime flare-corrupted images. As for flare-free images, we add light source annotations on flare-free images. The same operation is implemented on the BracketFlare dataset.

To improve the robustness of trained networks, we further introduce a data augmentation strategy. Specifically, to allow the network to perform better in nighttime images, we alter the gamma of flare-corrupted images by recovering the linear luminance of flare images and flare-free images with an inverse gamma correction strategy ($\gamma \sim U(1.8, 2.2)$). We also randomly multiply the RGB values with $U(0.5, 1.2)$ and add a Gaussian noise with variance sampled from a scaled chi-square distribution $\sigma^2 \sim 0.01\chi^2$. Furthermore, we use horizontal and vertical flips to enlarge the training set. Nevertheless, to enhance the network's abilities to deal with diverse flare, we carry out a series of affine transformations on flare images. The random blur on flare images with the kernel size in $U(0.1, 3)$ and offsets in $U(-0.02, 0.02)$ are also performed to control the brightness of flare images.

For the testing set, we adopt the real testing set provided by Flare7K++ and BracketFlare as the Flare7K++ real set mainly contains scattering flare and BracketFlare focuses on reflective flare. We conduct PIP pipeline on them for ablation experiments and comparison experiments.

### 4.2 IMPLEMENTATION DETAILS

Following Dai et al. Dai et al. (2022), we adopt PSNR, SSIM, and LPIPS as our metrics. Specifically, PSNR and SSIM evaluate the average pixel distance and the structure similarity between flare-corrupted images and flare-free images. Furthermore, LPIPS evaluates the feature-level distance between them. We set 4 levels in the PIN network. The channel of the immediate layer is 16. The optimizer for the PIP pipeline adopts the same setting as the FF-Former Zhang et al. (2023).

### 4.3 ABLATION RESULTS

In this section, we explore the function of the prompt calibration block and whether our pipeline is model-agnostic. Particularly, we do not conduct ablation experiments on our mask block. Without the mask block, our PIP pipeline will become a stacked U-net network. The pipeline will achieve better results by a deeper network with no doubt, whereas fails to suppress artifacts.

Table 1: Comparison of methods with or without prompt calibration block (PCB)

| Method | PSNR ↑ | SSIM ↑ | LPIPS ↓ |
|--------|--------|--------|---------|
| *w/o* PCB | 28.14 | 0.902 | 0.041 |
| *w* PCB | **28.44** | **0.906** | **0.039** |

Table 2: The number of parameters and Flops (512×512) with or without Prompt Inpainting Pipeline (PIP)

| Param(M)/Flops(G) | Unet | Uformer | FF-Former |
|-------------------|------|---------|-----------|
| *w/o* PIP | 9.0/62.36 | 20.4/162.1 | 46.5/407.8 |
| *w* PIP | 11.8/84.9 | 23.6/194.3 | 49.6/440.0 |

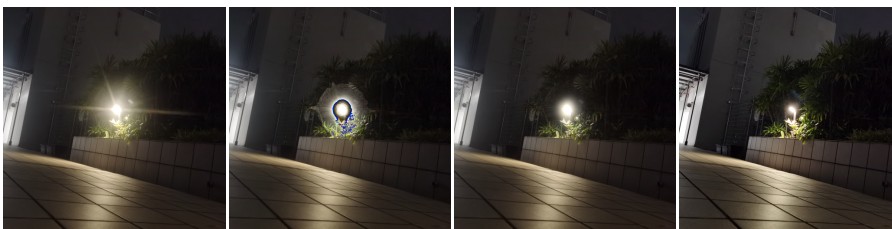

| Input | FF-Former + ZITS | FF-Former + PIP (Ours) | Ground Truth |

Figure 3: Visual comparison results on PIP pipeline VS inpainting network ZITS Dong et al. (2022). Severe artifacts happen in the image generated by FF-Former + ZITS.

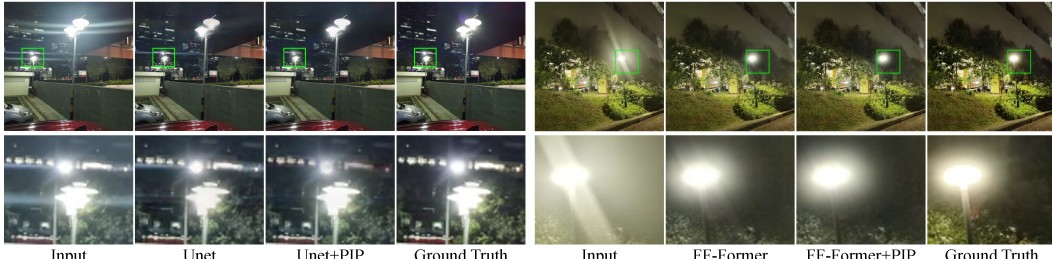

| Input | Unet | Unet+PIP | Ground Truth | Input | FF-Former | FF-Former+PIP | Ground Truth |

Figure 4: Visual comparison results of ablation experiments on PIP pipeline. We choose different images to better illustrate the function of our PIP as different network has weaknesses in different scenarios. Our PIP manages to further suppress artifacts, especially with a weaker network (Unet).

### 4.3.1 PROMPT CALIBRATION BLOCK

To verify the function of the proposed prompt calibration block and whether the visual prompt is necessary, we test how the pipeline will perform when no prompt calibration block are attended or other inpainting methods participate. To be fair, we adopt FF-Former as the base network in the coarse flare removal stage and test on the flare7k++ real set.

We show the comparison results on networks with or without prompt calibration block on Table 1. From the observation of Table 1, the network with the prompt calibration block surpasses the network without the prompt calibration block 0.3dB on PSNR, which proves that the visual prompt is non-substitutable and the prompt calibration block can provide significant PSNR score gains.

To further test whether other inpainting method can be useful for the flare removal task, we also compare it with ZITS Dong et al. (2022) method and adopt the official code for inference. The visual performance of the comparison is shown in Figure 3. Severe artifacts happen in the image generated by ZITS as the mask area is too complicated and uncertain to recover when much information has been discarded by the simple mask operation and no visual prompt can be used.

### 4.3.2 MODEL AGNOSTIC

To demonstrate that our PIP is model-agnostic and can be applied with any other flare removal network, we conduct quantitative and qualitative ablation experiments on Unet Ronneberger et al. (2015), Uformer Wang et al. (2022), and FF-Former Zhang et al. (2023). The network in the coarse flare removal stage is pretrained for predicting the coarse flare-free image and the flare image on Flare7K++ Dai et al. (2023a) dataset.

For quantitative comparison, based on Table 3, we can observe that with PIP pipeline, Unet Ronneberger et al. (2015) gains 0.92dB, 0.007, 0.004 on PSNR, SSIM, and LPIPS scores which surpass the state-of-the-art method FF-Former with 0.21dB, 0.001 on PSNR and SSIM score and achieve the same LPIPS score with it. Moreover, our PIP pipeline helps Uformer with 0.61dB, 0.006, and 0.003 gains on PSNR, SSIM, and LPIPS scores. As for the state-of-the-art method FF-Former Zhang et al. (2023), eqipped with our pipeline, it manages to obtain new state-of-the-art results, which achieve 28.44, 0.906, and 0.039 on PSNR, SSIM, and LPIPS scores and our pipeline provides 0.54, 0.006, and 0.002 promotions with a very small increase in the number of parameters, as shown in Table 2.

As for qualitative comparison, we show the visual comparisons on Figure 4. Compared with the original Unet, Unet + PIP pipeline removes more shimmer within the flare-coorupted images. As

Table 3: Comparison of flare removal methods on Flare7k++ real set and BracKetFlare dataset.

| Method | Flare7K | | | BracKetflare | | |
|---|---|---|---|---|---|---|
| | PSNR ↑ | SSIM ↑ | LPIPS ↓ | PSNR ↑ | SSIM ↑ | LPIPS ↓ |
| Wu et al. Wu et al. (2021) | 24.61 | 0.871 | 0.060 | 26.13 | 0.895 | 0.055 |
| Unet Ronneberger et al. (2015) | 27.19 | 0.894 | 0.045 | 47.00 | 0.897 | 0.006 |
| HINet Chen et al. (2021) | 27.55 | 0.892 | 0.046 | 48.03 | 0.994 | 0.003 |
| MPRNet* Zamir et al. (2021) | 27.04 | 0.893 | 0.048 | 48.41 | 0.994 | 0.004 |
| Restormer* Zamir et al. (2022) | 27.60 | 0.897 | 0.045 | 48.11 | 0.994 | 0.004 |
| Uformer Wang et al. (2022) | 27.63 | 0.894 | 0.043 | 47.47 | 0.991 | 0.003 |
| FF-Former Zhang et al. (2023) | 27.90 | 0.900 | 0.041 | 49.03 | 0.992 | 0.003 |
| Unet + PIP (Ours) | 28.11 | 0.901 | 0.041 | 48.50 | 0.992 | 0.004 |
| Uformer + PIP (Ours) | 28.24 | 0.903 | 0.040 | 48.69 | 0.994 | 0.002 |
| FF-former + PIP (Ours) | **28.44** | **0.906** | **0.039** | **49.18** | **0.994** | **0.002** |

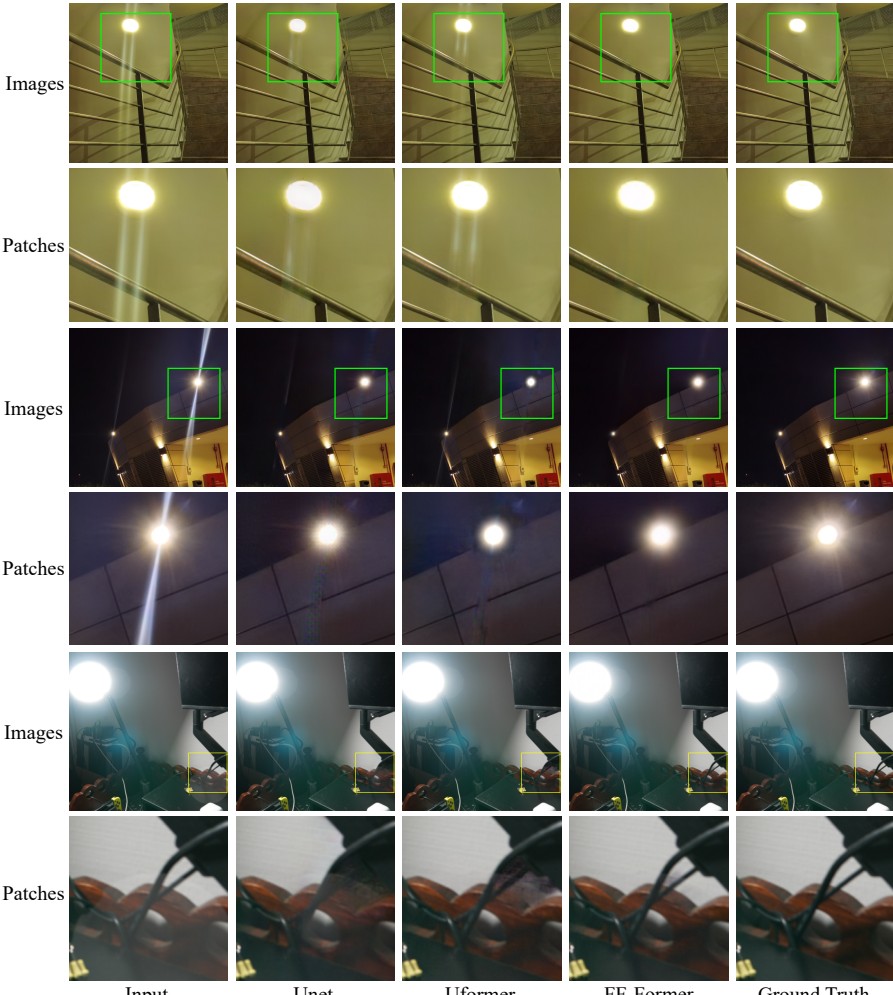

Figure 5: Visual comparison of the scattering flare and the reflective flare. Images in the first four rows are from the flare7k++ real set and mainly contain scattering flare. Images in the last two rows are from the BraceketFlare dataset and mainly contain reflective flare. Our method eliminates the most flare and generates images with the fewest artifacts.

for FF-Former, more streaks have been eliminated. Previous pipelines hardly identify and eliminate all the flares added to the image and tend to generate severe artifacts when handling strong flares with large overexposed streaks and shimmers. Our PIP pipeline suppresses this phenomenon by rewriting the image details polluted by flare with the extracted semantic information within the unpolluted area and the multi-scale features obtained in the coarse flare removal stage. The experiments demonstrate that our pipeline is model-agnostic.

### 4.4 COMPARISON RESULTS

In this section, we conduct comparison results on the real testing set of the Flare7K++ dataset and BracketFlare dataset to demonstrate the functions of our PIP pipeline. We employ flare removal methods Wu et al. Wu et al. (2021) and FF-Former Zhang et al. (2023) and image restoration methods Unet Ronneberger et al. (2015), HINet Chen et al. (2021), MPRNet Zamir et al. (2021), Restormer Zamir et al. (2022), and Uformer Wang et al. (2022) for baselines.

#### 4.4.1 RESULTS ON FLARE7K++ DATASETS

To validate the performance of our method on complex scenarios which include both the scattering flare and reflective flare, we show the quantitative results (*e.g.*, PSNR, SSIM, LPIPS) on Table 3 for experiments on Flare7k++ real testing set, correspondingly. Furthermore, we also represent the visual comparison results on Figure 5 for experiments on the Flare7k++ real testing set (we employ FF-Former Zhang et al. (2023) in coarse flare removal network).

From the observation of Table 3, our PIP pipeline equipped with FF-Former Zhang et al. (2023) has achieved state-of-the-art PSNR and SSIM. Concretely, from the perspective of pixel level, we significantly outperform the state-of-the-art method FF-Former with 0.54dB on PSNR score, 0.006 on SSIM, and 0.002 on LPIPS. As for visual performance comparison, based on the first four rows in Figure 5, we figure that methods trained in a supervised manner achieve better results and our PIP pipeline equipped with FF-Former has essentially eliminated the flare and introduced the least artifacts in most situation. Despite the shape of the flare, round flare with a large radius, flare with a long streak, or the type of the flare, reflective flare, or scattering flare, our method can achieve more realistic and natural results than other baselines in most scenarios.

#### 4.4.2 RESULTS ON BRACKETFLARE DATASETS

To further estimate our method on reflective flare, we test our method on the BracKetFlare dataset Dai et al. (2023b). From the observation of Table 3, our PIP pipeline equipped with FF-Former surpasses MPRNet with 0.77dB on PSNR and 0.002 on LPIPS and obtains the same SSIM. Furthermore, based on the last two rows in Figure 5, we significantly outperform MPRNet by thoroughly eliminating the reflective flare with the fewest artifacts, which proves that some artifacts may occur during the reflective flare removal process and our PIP manages to depress such phenomenon.

## 5 DISCUSSION

To suppress the artifacts created during the flare removal process, we propose the two-stage pipeline named PIP. The PIP pipeline borrows the idea from the inpainting task and adopts a mask block to construct the input of the inpainting neural network. However, as shown in Figure 3, simply adopting the inpainting method will lead to essential content loss as plenty of the image content is hidden by the flare, which leads the current inpainting method hardly recovers them and the extracted flare mask is error-prone, which makes the inpainting method mistakenly rewrites the flare instead of image content. Therefore, we adjust the inpainting method for the flare removal task. By adopting the features extracted from the coarse flare removal stage as inference, the prompt calibration block in the image refinement stage manages to rewrite the blocked image content.

## 6 CONCLUSION

In this paper, we propose a model-agnostic two-stage pipeline named PIP pipeline. By giving the flare removal task a second thought, the PIP pipeline provides a novel view for the flare removal task by rewriting the pixels corrupted by the flare. The PIP pipeline consists of a coarse flare removal stage and an image refinement stage. The coarse flare removal stage generates coarse flare-free images and the estimated flare image and the image refinement stage alleviates the artifacts by adopting the idea of inpainting methods. However, instead of compelling the network to infer the masked image, we accomplish the mission by extracting the semantic information of the unmasked area and employing multi-scale features as visual prompts. Extensive results on real-world and synthetic benchmarks demonstrate the superiority of the PIP pipeline.

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
