# OpenReview forum: "Flare Removal with Visual Prompt"
_ICLR.cc/2025/Conference — ICLR 2025 Conference Withdrawn Submission_

### Official Review · Reviewer_vUoL · 2024-10-19

**Soundness:** 2
**Presentation:** 3
**Contribution:** 2
**Rating:** 3
**Confidence:** 4

**Summary:**

This paper focuses on the flare removal task. A model-agnostic pipeline, i.e., Prompt Inpainting Pipeline (PIP), is proposed to remove the flare by masking the polluted area and rewriting image details. It consists of a coarse flare removal stage and an image refinement stage. A prompt calibration block is further proposed to guide the rewriting process. The experimental results demonstrate the effectiveness of the proposed method.

**Strengths:**

The proposed method is model-agnostic, and can be applied to existing Unet-based flare removal methods. Applying the inpainting idea to the polluted area is a reasonable solution to improve the flare removal quality. The experiments show the superiority of the proposed method.

**Weaknesses:**

The main weaknesses are listed as follows.

1. The key idea of this paper is to follow the inpainting method to mask the polluted area and rewrite image details within. However, such an idea is not new, and can be used in general low-level vision tasks like reflection or shadow removal. I cannot see new specific insights for the flare removal task from this paper.

2. The technical contribution of this paper is not very clear. The whole pipeline can be considered as an incremental work with two stages. The first coarse flare removal stage belongs to existing Unet-based flare removal methods. The second image refinement stage is another Unet to apply the inpainting idea. The SFB blocks are also from an existing work.

3. As this paper can be considered as adding an inpainting module to the existing flare removal methods, a detailed computation cost and running time analysis are necessary.

4. The current visual results are a little bit confusing. I am not sure about the correspondence between the results in Fig.5 and Table 3. There are 10 rows in Table 3, but only 5 columns in Fig.5. In addition, take the 4th row as an example. It seems that the quality of the ground truth image in the last column is even worse than that of the images in other columns.

5. The ablation study is not thorough. The effects of the data augmentation strategy and the positions of the PCB have not been studied yet. Can PCB be used in the encoder?

**Questions:**

1. Can ZITS represent the SOTA performance for the image inpainting task? I was wondering about the flare removal results if the FF-former+ZITS pipeline is finetuned on the training dataset used by the proposed method.

2. Since the flare mask is obtained from the coarse flare removal stage, what if the estimated mask is incorrect?

---

### Official Review · Reviewer_d6LH · 2024-10-27

**Soundness:** 2
**Presentation:** 3
**Contribution:** 2
**Rating:** 3
**Confidence:** 5

**Summary:**

This paper propose a new pipeline namely prompt inpainting pipeline (PIP) for flare removal task. The proposed PIP is mainly designed basing on the baseline of Flare 7k++, that additionally introduces a image inpainting sub-network to further improve the performance of baseline. Through the experiments, authors demonstrate that the proposed PIP could improve the performance of multiple baselines.

**Strengths:**

1. This paper is well organized and easy to follow.
2. Experiment results demonstrate that the proposed pipeline could improve the performance of multiple baselines.

**Weaknesses:**

1. The motivation of this work is unclear. Authors argue (in line 053) that removing flare from images without introducing artifacts is challenging. However it's hard to build a relationship between this challenge and the proposed pipeline.

2. The major problem of this paper is that the novelty is quite limited. Introducing inpainting methods to work associate the spatial mask has been widely used in various image processing methods. Besides, the visual prompt and calibration block have also been proposed.

3. The proposed PIP seems to be supposed to improve the baseline by avoiding introducing artifacts during removing the flare. However, the provided visualization results cannot clearly support this point.

**Questions:**

See weakness.

---

### Official Review · Reviewer_7cAK · 2024-10-31

**Soundness:** 2
**Presentation:** 1
**Contribution:** 2
**Rating:** 5
**Confidence:** 5

**Summary:**

This paper tackles the flare removal problem by proposing a two-stage prompt inpainting pipeline. The approach includes a coarse flare removal stage followed by an image refinement stage. In the first stage, a U-shaped network is used to generate a coarse flare-free image and the flare itself. During the refinement stage, a prompt calibration block utilizes features extracted from the initial stage as a visual prompt to achieve refined, flare-free results.

**Strengths:**

This paper proposes a two-stage pipeline to address the flare removal problem.
The quantitative results seem good.

**Weaknesses:**

1.	The results in Figure 3 lack validity; the ZITS model should be retrained on the flare dataset. This is crucial for the paper’s argument and requires careful consideration.
The presentation is poor.
2.	The Equations are hard to follow; for instance, symbols like F_{DF} in Eq.1 are overly casual.
3.	Some sentences are unclear or inaccurate. For example, in lines 243-245: “To this end, we accomplish this task from a novel view by extracting the semantic information from the unpolluted area, rewriting the details in the polluted area based on the extracted semantic information, and using multi-scale features extracted from the last stage as the visual prompt.” The sentence structure may be confusing for readers.
4.	The paper should be reorganized; for instance, sections 4.3.2 and 4.4 contain overlaps.
5.	Statements in lines 452-454 lack sufficient support or evidence.
6.	In lines 460-462, MPRNet is metioned. However, Figure 5 appears to be missing MPRNet.

**Questions:**

1.	Eq.1 is unclear. What’s the meaning of F_{gt}^i in this equation.
2.	In Table 2, why do the parameter and FLOP gains with PIP vary for different methods?

---

### Official Review · Reviewer_7uNa · 2024-11-08

**Soundness:** 3
**Presentation:** 3
**Contribution:** 3
**Rating:** 5
**Confidence:** 5

**Summary:**

The paper proposes a model-agnostic pipeline named PIP that effectively suppresses artifacts by rewriting details that have been corrupted by flare. The proposed prompt calibration block utilizes features extracted during the coarse flare removal stage as a visual prompt to guide the rewriting process. Extensive experiments are conducted on real-world benchmarks, demonstrating the effectiveness and superiority of the proposed method.

**Strengths:**

This paper proposes a new method for removing flare by combining the estimated flare image with a coarse flare-free image using an additional network. By utilizing only 10% more computational resources, the method achieves better results.
The writing of the paper is relatively clear.

**Weaknesses:**

This paper addresses the flare removal problem using a two-stage approach with an increase of 10% in computational resources. The experiments primarily focus on solving the issue by adding modules to Uformer and FF-Former. However, the approach is quite similar to that of Multi-Stage Progressive Image Restoration (CVPR 2021). The paper should compare its results with other similar multi-stage restoration papers in the experiments. If relevant citations and experiments are included, I will give a positive score.

**Questions:**

see above

---

### Note · Authors · 2024-11-13

I have read and agree with the venue's withdrawal policy on behalf of myself and my co-authors.